# Landau-Lifshitz damping from Lindbladian dissipation in quantum magnets

Götz S. Uhrig[1, *]

[1]*Condensed Matter Theory, Department of Physics,*
*TU Dortmund University, Otto-Hahn-Straße 4, 44221 Dortmund, Germany*

(Dated: June 18, 2024)

As of now, the phenomenological classical Landau-Lifshitz (LL) damping of magnetic order is not linked to the established quantum theory of dissipation based on the Lindbladian master equation. This is an unsatisfactory conceptual caveat for the booming research on magnetic dynamics. Here, it is shown that LL dynamics can be systematically derived from Lindbladian dynamics. Thereby, the successful LL approach is set generally on a firm quantum basis. The key aspect is that the Lindbladian relaxation must be adapted to the Hamiltonian $H(t)$ at each instant of time in time-dependent non-equilibrium systems. It is conjectured that this idea holds true well beyond the damping of magnetic dynamics.

Magnetic dynamics is a topic attracting an enormous interest these days because it is crucial for making progress in magnonics [1], i.e., for the general idea to use magnetic degrees of freedom for information storage and handling. We stress that this requires to deal with non-equilibrium physics. Ferromagnetic [2, 3] and antiferromagnetic systems [4, 5] are considered with their respective advantages and disadvantages. For substantial progress numerous experimental challenges have to be overcome and well-founded theoretical understanding is indispensable. This requires to capture the relevant Hamiltonian dynamics as well as the relaxation processes. While the former is often known from experimental measurements or even from *ab initio* calculations, the latter is generically described phenomenologically on a classcial level by the Landau-Lifshitz (LL) equation or the Landau-Lifshitz-Gilbert (LLG) equation [6–8]. These equations, though phenomenological, are fundamental to realistic descriptions of magnetism off equilibrium.

Since spins as the elementary objects of magnetism are quantum entities and in view of the importance of the LL(G) equations it is not surprising that there are intensive efforts to establish a quantum foundation for them [9–12]. One key idea is to establish a quantum mechanical equation that results in the established classical equations in the classical limit [9, 10, 12]. The assumptions necessary to achieve this goal are a strong non-Hermitian Hamiltonian [9, 10] or a self-referential term in the von-Neumann equation where the density matrix appears twice [12]. Further studies are called for justifying these assumptions on the basis of physical processes. In parallel, a number of approaches there exists which compute the Gilbert damping parameter obtaining reasonable values on the basis of time-dependent or non-equilibrium quantum mechanical approaches [11, 13–20]. This suggests that it must be possible to derive the damping in the LL(G) equations from standard quantum mechanics. It is the objective of this paper to provide such a derivation.

There exists a well-established and successful theory for relaxation of quantum systems. The key foundation is to consider an open quantum system coupled to a bath [21] leading to the Lindblad equation including a so-called dissipator. If the damping of a magnetic systems can be described by the LL(G) equations, the latter must be derivable from Lindbladian dynamics. So far, however, Lindblad dynamics and LL(G) dynamics seem to stand completely separate without a logical link. This situation is highly unsatisfactory since the spins are quantum objects and cannot have their own relaxation independent from the general quantum theory. In this study, we will fill this obvious gap and link Lindbladian dynamics and LL(G) dynamics. We show that the latter results from the former in a very general limit.

As a brief recapitulation, the LL equation reads

$$\frac{d\hat{\vec{m}}}{dt} = \hat{\vec{m}} \times \vec{h}_0 - \lambda \hat{\vec{m}} \times (\hat{\vec{m}} \times \vec{h}_0) \tag{1a}$$

if the Hamilton function reads $H = -\vec{h}_0 \cdot \hat{\vec{m}}$ and $\hat{\vec{m}}$ is the normalized magnetization of a spin and $\vec{h}_0$ a fixed external magnetic field with $h_0 = g\mu_{\mathrm{B}}B$. The LLG equation reads

$$\frac{d\hat{\vec{m}}}{dt} = \hat{\vec{m}} \times \vec{h}_0 - \lambda \hat{\vec{m}} \times \frac{d\hat{\vec{m}}}{dt}. \tag{1b}$$

Both equations are equivalent within $\mathcal{O}(\lambda^2)$ [10, 22] because (1b) can be transformed to

$$\frac{d\hat{\vec{m}}}{dt} = \frac{1}{1+\lambda^2} \left[ \hat{\vec{m}} \times \vec{h}_0 - \lambda \hat{\vec{m}} \times (\hat{\vec{m}} \times \vec{h}_0) \right]. \tag{2}$$

using the following property of the cross product with a unit vector $\hat{\vec{m}}$. Applying $\hat{\vec{m}} \times$ to a vector in $\mathbb{R}^3$ defines a linear mapping $K : \mathbb{R}^3 \rightarrow \mathbb{R}^3$ fulfilling $K^3 = -K$ as can be shown easily by the so-called bac-cab rule $a \times (b \times c) = b(a \cdot c) - c(a \cdot b)$ for $a, b, c \in \mathbb{R}^3$. Then the inverse $(1 + \lambda K)^{-1}$ is given by $(1 + \lambda^2 - \lambda K + \lambda^2 K^2)/(1 + \lambda^2)$ as can be checked by direct evaluation. Hence the LL

and LLG equation stand for almost the same dynamics except for a renormalization of the time $t \to (1 + \lambda^2)t$.

Lindbladian dynamics for an operator $A$ in the Heisenberg picture is given by the adjoint Master equation which reads at zero temperature [21, 23]

$$\frac{d\langle A \rangle}{dt} = i\langle [H, A] \rangle + \frac{1}{2} \sum_l \gamma_l \left\langle [B_l, A] B_l^\dagger + B_l [A, B_l^\dagger] \right\rangle \quad (3)$$

where the $\gamma_l$ are the decay rates of the various dissipative channels which are defined by the Lindblad operators $B_l$. It is implied that $B_l$ *increments* the energy in $H$ by $\hbar\omega_l > 0$ while $B_l^\dagger$ decreases the energy by the same amount [24].

Considering a single spin coupled to a magnetic field which we assume for simplicity to point into $z$ direction $\vec{h}_0 = h_0(0, 0, 1)^\top$) the minimal relaxation channel is given by $B = S^+$ with rate $\gamma$. Then, Eq. (3) implies

$$\frac{d\langle S^z \rangle}{dt} = \gamma(S - \langle S^z \rangle) \quad (4a)$$

$$\frac{d\langle S^+ \rangle}{dt} = -(ih_0 + \gamma S)\langle S^+ \rangle \quad (4b)$$

with the spin $S$. These equations are exact for $S = 1/2$; for general spin they are valid if the spin state deviates only little from the state in the polarized $z$ direction, i.e., only the magnetic quantum numbers $S$ and $S - 1$ occur. We will discuss the justification of this approximation below. The solutions of (4) read $\langle S^z \rangle(t) = (\langle S^z \rangle(0) - S) \exp(-\gamma t) + S$ and $\langle S^+ \rangle(t) = \exp(-(ih_0 + \gamma S)t)\langle S^+ \rangle(0)$. This reflects standard longitudinal relaxation combined with damped transverse precession, i.e., the physically expected result. Yet, this behavior is strikingly different from what follows from Eqs. (1)! But this is not the end of the story.

As an important intermediate step we consider the magnetic field $\vec{h}$ to be time dependent; its origin will be discussed below. Then, the Lindbladian dynamics (4) adapts to $\vec{h}(t)$ at each instant of time. For the total expectation value $\vec{m} ::= \langle \vec{S} \rangle$ we obtain after some algebra

$$\frac{d\vec{m}}{dt} = \vec{m} \times \vec{h} + \gamma(S\vec{n} - \vec{m}) - \gamma(1 - S)\vec{n} \times (\vec{n} \times \vec{m}), \quad (5)$$

where $\vec{n} = \vec{h}/h$ is the time-dependent unit vector in the direction of the magnetic field [25]. In the first term on the right hand side one recognizes the precession and the last term is a double cross product, but twice with $\vec{n}$ instead of $\hat{\vec{m}}$. The self-reference is missing which one has in Eqs. (1).

Here, the next key element enters. We do not attribute the self-reference to an assumed modification of the Schrödinger equation, but to a mean-field treatment considering the *ensemble* of spins. For simplicity, we

treat a translational invariant ferromagnet in which each spin is subject to the external field $\vec{h}_0 = h_0(0, 0, 1)^\top)$ and to the internal exchange field generated by the surrounding spins interacting with the spin under study so that

$$\vec{h} = \vec{h}_0 + J\vec{m}, \quad (6)$$

where $J$ represents the sum over all exchange couplings acting on the considered spin. Inserting (6) into (5) yields a closed, self-referential set of equations for the magnetization $\vec{m}$. Still, it seems to be distinctly different from Eqs. (1), but this is illusive. A first step to see this is to estimate orders of magnitude. Generic ferromagnets have a Curie temperature of about $10^3$ K corresponding roughly to an internal magnetic field of $10^3$ T. Externally applied fields range from 1 mT to 10 T. Thus, we face the situation that $h_0$ is about (more than) 1000 times weaker than the internal field $JS$.

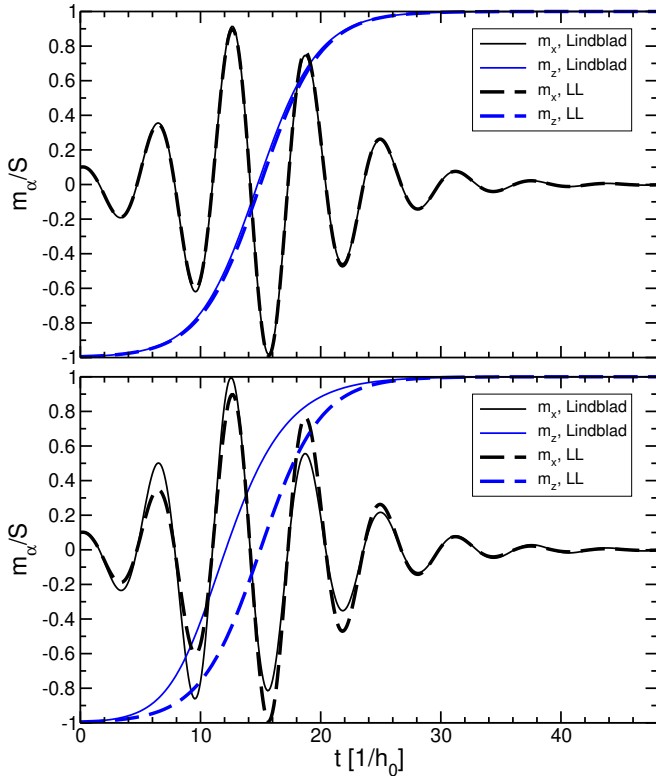

FIG. 1. Solutions (dashed lines) of the Landau-Lifshitz equation (1a) with $h_0 = 1$ as energy unit (magnetic field in $z$ direction) and $\lambda = 0.2$. Solutions (solid lines) of the Lindbladian equation (5) at $S = 1/2$ for the same parameters and $J = 100h_0$ (upper panel) as well as $J = 8h_0$ (lower panel). The initial direction of the magnetization is tilted by $\pi - 0.1$ with respect to the $z$ direction and has the length $S$. Note the excellent agreement in the upper panel and the increased deviation in the lower panel although the qualitative behavior is still the same.

Figure 1 confirms the resulting conjecture that the LL dynamics is the weak-field limit of the Lindbladian dy-

namics. The dashed lines show the solutions of Eq. (1a) while the solid lines in the same color represent the solutions of Eq. (5) using (6). Clearly, for $J/h_0 = 100$ the agreement is excellent; only a minor shift in the relaxation of the $z$ component is discernible. Lowering the ratio $J/h_0$ the deviations increase, but the qualitative behavior still is very close to the LL dynamics. Figure 2 shows that for significant lower ratios the qualitative behavior changes and $z$ component essentially displays exponential relaxation as one would expect from (4). At this point, we emphasize that the use of Eq. (6) for large ratios $J/h_0$ implies that the spin orientation is always very close to the direction of the effective magnetic field $\vec{h}$, not to $\vec{h}_0$. This justifies the use of the approximation necessary to establish (4) for $S > 1/2$. For $S = 1/2$, the differential equations (4) are exact anyway.

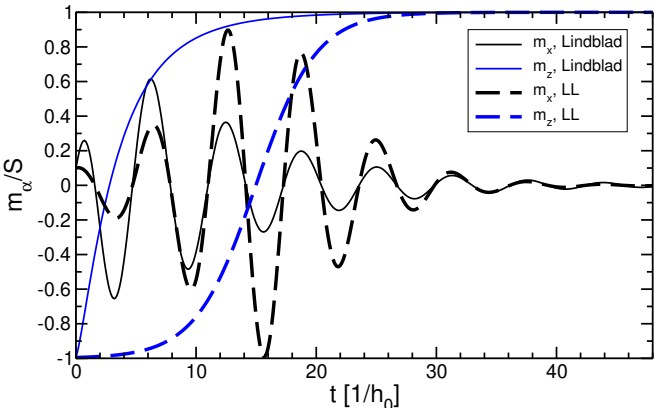

FIG. 2. Same as in Fig. 1 except for $J = 2h_0$. Note the qualitatively different behavior in particular of $m_z$ resembling exponential relaxation rather than Landau-Lifshitz damping.

The numerical agreement can be backed by analytical arguments [25]. Expanding the right hand side (5) in linear order in $h_0$ using $\vec{n} = \vec{h}/h$ and (6) yields

$$\frac{1}{S}\frac{d\vec{m}}{dt} = \frac{\vec{m}}{S} \times \vec{h}_0 + \gamma(\hat{\vec{m}} - \vec{m}/S) - \frac{\gamma}{J}C\,\hat{\vec{m}} \times (\hat{\vec{m}} \times \vec{h}_0). \quad (7)$$

with $C := 1 + 1/|\vec{m}| - 1/S$. Note that we distinguish here between the magnetization with arbitrary length $\vec{m}$ and the normalized one $\hat{\vec{m}}$. Inspecting the three terms on the right hand side of (7) the first one induces the precession. It is the fastest for $h_0 > \gamma > h_0\gamma/J$. The second one is also a fast one; it could even be faster than the precession depending on the applied field. Clearly, it is faster than the third one since we are considering the regime of large $J$. The second term ensures that the magnetization converges to its saturation value $S$ quickly on the time scale $1/\gamma$. If we plausibly start from an initial magnetization that takes its saturation value this term has no effect anymore and $|\vec{m}| = S$ holds for all subsequent times. This implies $C = 1$ and that the time derivative on the left hand side equals $\frac{d\hat{\vec{m}}}{dt}$ as well as the

precession term equals $\hat{\vec{m}} \times \vec{h}_0$. The third term on the right hand side represents the LL damping term if we set $\lambda = \gamma/J$.

Thus, we eventually derived the LL equation (1a) from Lindbladian dynamics and identified the dimensionless damping parameter $\lambda$ by the ratio of a relaxation rate $\gamma$ to the internal energy scale $J$. The latter should be larger than $\gamma$ in order that the standard assumption for deriving Lindbladian dynamics hold [21]. Hence we can safely assume $\lambda \leq 1$ so that no unphysical behavior results from the LL equation. In addition, we studied larger values of $\lambda$ to see whether the LL equation (1a) or the LLG equation (1b) corresponds better to the Lindbladian dynamics (5) [25]. In accordance with the analytical derivation of the weak-field limit we find that the LL equation represents the limit of weak external fields best.

In summary, up to now there was a conceptual gap between the established Lindbladian theory for dissipation in open quantum systems and the phenomenological theory of Landau, Lifshitz and Gilbert for the damping of the temporal behavior of magnetic order. In view of the humongous interest in magnetic dynamics this represented a serious flaw. So far, the differences of the two approaches were emphasized, for instance the change of entropy in Lindblad treatments, see e.g. Ref. 26, and the conservation of entropy in the assumed quantum analogues of the LL(G) equations [9, 12]. In the present study, we filled the conceptual gap and reconciled the fundamental quantum description by means of a Lindbladian dissipator and the phenomenological classical LL(G) description. Three key points are important: (i) the Lindbladian relaxation is constantly adapted to the instantaneous Hamilton operator and its energies, i.e., the relaxation always acts such as to minimize the instantaneous energy; (ii) the self-reference in the LL(G) equations is explained not by ad hoc modifications of the Schrödinger equation, but by a standard mean-field treatment; (iii) finally the internal energy scales of the quantum system are generically much larger than the external ones, here $|J| \gg |h_0|$. These ideas allowed us to derive the Landau-Lifshitz equation analytically from Lindbladian dynamics and to confirm this by numerical calculations. We did not observe a trace of Landau-Lifshitz-Gilbert dynamics for large values of $\lambda$.

The possibility to derive LL dynamics from Lindbladian dynamics provides the general context for computing damping parameters from elaborate theories describing the reservoirs explicitly which are responsible for the relaxation of quantum system [11, 13–20]. The above ideas can also be applied directly to quantum magnets beyond ferromagnets as along as they are treated by a mean-field approach. An important example are quantum antiferromagnets gaining more and more interest for the prospect of faster data handling [4, 27]. Quantum fluctuations,

however, are not captured [28, 29] by approaches relying on macrospins only. But the success of the above treatment clearly suggests that Lindbladian dissipation captures the physics of relaxation well. The above key idea (i) that it must be constantly adapted to the instantaneous Hamiltonian in time-dependent, non-equilibrium situations is generally applicable, thus paving the way to studies of relaxation in a plethora of non-equilibrium quantum systems.

The author is thankful to Asliddin Khudoyberdiev for useful discussions. This research has been supported by the Deutsche Forschungsgemeinschaft (DFG) in project UH 90-14/1 and by the Stiftung Mercator in project KO-2021-0027.

---

* goetz.uhrig@tu-dortmund.de

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

## SUPPLEMENTAL MATERIAL

### Derivation of Lindbladian dynamics for time-dependent magnetic field

The aim is to derive Eq. (5). We choose $\vec{n} := \vec{h}/h$ and an orthogonal unit vectors $\vec{p}$ so that they for a orthonormal basis of $\mathbb{R}^3$ together with $\vec{q} := \vec{n} \times \vec{p}$. Their completeness is expressed by

$$\vec{n}\vec{n}^\top + \vec{p}\vec{p}^\top + \vec{q}\vec{q}^\top = \mathbb{1}_3 \qquad (8)$$

where $\mathbb{1}_3$ is the $3 \times 3$ identity matrix. Differentiation of the completeness with respect to time yields

$$\frac{d\vec{n}}{dt}\vec{n}^\top + \vec{n}\frac{d\vec{n}}{dt}^\top + \frac{d\vec{p}}{dt}\vec{p}^\top + \vec{p}\frac{d\vec{p}}{dt}^\top + \frac{d\vec{q}}{dt}\vec{q}^\top + \vec{q}\frac{d\vec{q}}{dt}^\top = 0 \quad (9)$$

which we will use below. For a concise notation we introduce the complex vector

$$\vec{z} := \vec{p} + i\vec{q} \in \mathbb{C}^3. \qquad (10)$$

Then we can compute the longitudinal and transverse spin expectation value using $\vec{m} = \langle \vec{S} \rangle$

$$S_\parallel := \vec{n} \cdot \vec{m} \qquad (11a)$$
$$S_+ := \vec{z} \cdot \vec{m}. \qquad (11b)$$

These values acquire their time dependence from Eqs. (4) and from the time dependence of $\vec{n}$ and $\vec{z}$

$$\frac{dS_\parallel}{dt} = \gamma(S - S_\parallel) + \frac{d\vec{n}}{dt} \cdot \vec{m} \qquad (12a)$$
$$\frac{dS_+}{dt} = -(ih + \gamma S)S_+ + \frac{d\vec{z}}{dt} \cdot \vec{m}. \qquad (12b)$$

Reconstructing $\vec{m}$ from the two components defined in (11) yields

$$\vec{m} = \vec{n}S_\parallel + \frac{1}{2}\vec{p}(S_+ + S_+^*) - \frac{i}{2}\vec{q}(S_+ - S_+^*) \qquad (13a)$$
$$= \vec{n}S_\parallel + \Re(\vec{z}^* S_+). \qquad (13b)$$

Computing the derivative of $\vec{m}$ from (13b) and (12) yields

$$\frac{d\vec{m}}{dt} = \frac{d\vec{n}}{dt}(\vec{n} \cdot \vec{m}) + \vec{n}\left(\frac{d\vec{n}}{dt} \cdot \vec{m}\right) \qquad (14a)$$
$$+ \Re\left(\frac{d\vec{z}^*}{dt}(\vec{z} \cdot \vec{m}) + \vec{z}^*\left(\frac{d\vec{z}}{dt} \cdot \vec{m}\right)\right) \qquad (14b)$$
$$+ \gamma\vec{n}(S - \vec{n} \cdot \vec{m}) - \Re\left((ih + \gamma S)\vec{z}^*(\vec{z} \cdot \vec{m})\right). \quad (14c)$$

The lines (14a) and (14b) cancel due to (9). Using

$$\Re(\vec{z}^*\vec{z}^\top) = \vec{p}\vec{p}^\top + \vec{q}\vec{q}^\top = \mathbb{1}_3 - \vec{n}\vec{n}^\top, \qquad (15)$$

where the second equation results from (8), as well as

$$\Im(\vec{z}^*(\vec{z} \cdot \vec{m})) = \vec{p}(\vec{q} \cdot \vec{m}) - \vec{q}(\vec{p} \cdot \vec{m}) \qquad (16a)$$
$$= \vec{m} \times (\vec{p} \times \vec{q}) = \vec{m} \times \vec{n} \qquad (16b)$$

leads us to

$$\frac{d\vec{m}}{dt} = \vec{m} \times \vec{h} + \gamma S\vec{n} - \gamma(1 - S)\vec{n}(\vec{n} \cdot \vec{m}) - \gamma S\vec{m}. \quad (17)$$

To the right hand side we add and subtract $\gamma(1 - S)\vec{m}$ so that we can use $\vec{m} - \vec{n}(\vec{n} \cdot \vec{m}) = -\vec{n} \times (\vec{n} \times \vec{m})$ to obtain

$$\frac{d\vec{m}}{dt} = \vec{m} \times \vec{h} + \gamma(S\vec{n} - \vec{m}) - \gamma(1 - S)\vec{n} \times (\vec{n} \times \vec{m}). \quad (18)$$

This is the sought equation (5).

### Linear order in the external field $h_0$

The first term in (18) takes the form

$$\vec{m} \times \vec{h} = \vec{m} \times \vec{h}_0 \qquad (19)$$

because $\vec{m} \times \vec{m} = 0$; this is linear in $h_0$. In the second term, we have to use

$$h^2 = J^2|\vec{m}|^2 + 2J\vec{h}_0 \cdot Jvm + h_0^2 \qquad (20)$$

to reach

$$\vec{n} = \frac{\vec{h}}{h} \qquad (21a)$$
$$= \frac{\vec{h}_0}{J|\vec{m}|} + \hat{\vec{m}}\left(1 - \frac{\vec{h}_0 \cdot \hat{\vec{m}}}{J|\vec{m}|}\right) + \mathcal{O}(h_0^2) \qquad (21b)$$
$$= \hat{\vec{m}} + \frac{1}{J|\vec{m}|}(\vec{h}_0 - \hat{\vec{m}}(\hat{\vec{m}} \cdot \vec{h}_0)) + \mathcal{O}(h_0^2) \qquad (21c)$$
$$= \hat{\vec{m}} - \frac{1}{J|\vec{m}|}\hat{\vec{m}} \times (\hat{\vec{m}} \times \vec{h}_0) + \mathcal{O}(h_0^2) \qquad (21d)$$

where we use the normalized magnetization $\hat{\vec{m}} = \vec{m}/|\vec{m}|$. The third term in (18) simplifies according to

$$\vec{n} \times (\vec{n} \times \vec{m}) = \frac{\vec{h} \times (\vec{h} \times \vec{m})}{h^2} \qquad (22a)$$
$$= \frac{\vec{h} \times (\vec{h}_0 \times \vec{m})}{h^2} \qquad (22b)$$
$$= \frac{J\vec{m} \times (\vec{h}_0 \times \vec{m})}{J^2|\vec{m}|^2} + \mathcal{O}(h_0^2) \qquad (22c)$$
$$= -\frac{\hat{\vec{m}} \times (\hat{\vec{m}} \times \vec{h}_0)}{J} + \mathcal{O}(h_0^2). \qquad (22d)$$

Adding the results (19), (21d), and (22d) with the appropriate prefactors yields

$$\frac{d\vec{m}}{dt} = \vec{m}\times\vec{h}_0 + \gamma(S\hat{\vec{m}}-\vec{m}) - \frac{\gamma}{J}\Big(\frac{S}{|\vec{m}|}+S-1\Big)\hat{\vec{m}}\times(\hat{\vec{m}}\times\vec{h}_0).$$
(23)

Dividing by $S$ yields the wanted equation (7) given in the main text.

### Effect of large Landau-Lifshitz damping

The figures in the main text and the analytical calculation presented above in the previous section indicate that the LL equation (1a) captures the limit of weak external fields of the Lindbladian dynamics best. Yet, the damping value used in the numerical computation, $\lambda = 0.2$, is fairly small and any possible effect on the dynamics would be only 4%, see Eq. (2). In order to corroborate the observation that the Lindbladian dynamics reduces to LL dynamics rather than LLG dynamics we study the case $\lambda = 0.5$ in Fig. 3.

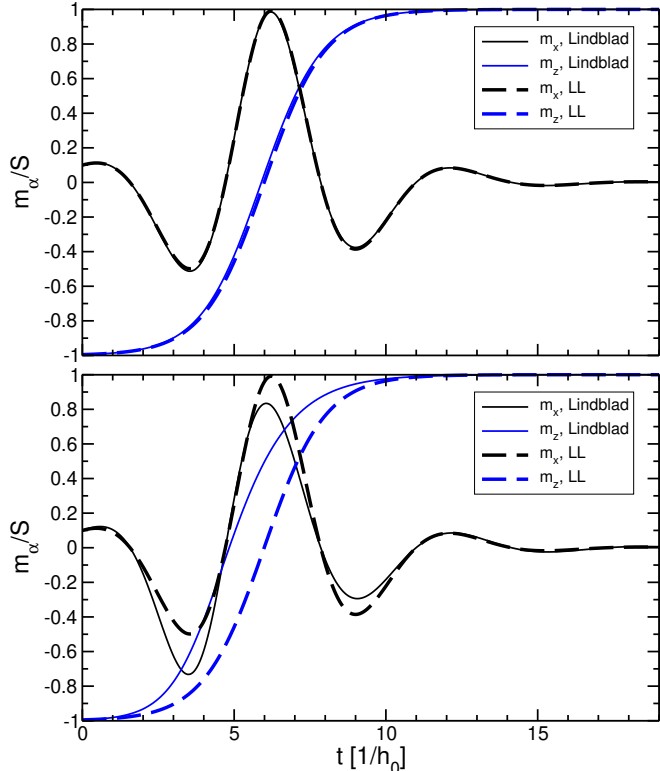

FIG. 3. Solutions (dashed lines) of the Landau-Lifshitz equation (1a) with $h_0 = 1$ as energy unit (magnetic field in $z$ direction) and $\lambda = 0.5$. Solutions (solid lines) of the Lindbladian equation (5) at $S = 1/2$ for the same parameters and $J = 100h_0$ (upper panel) as well as $J = 8h_0$ (lower panel). The initial direction of the magnetization is tilted by $\pi - 0.1$ with respect to the $z$ direction and has the length $S$.

Of course, the damping takes place faster by a factor of 2.5 than in Fig. 1. But we find again that the agreement for large $J$ in the upper panel is almost perfect while deviations become discernible for intermediate $J$ in the lower panel. But the qualitative behavior remains the same. We do not observe any shift in the oscillations of the $m_x$ component relative to the oscillations in the corresponding LL curve. Hence, we do not see any sign of a renormalization of the precession term. From the LLG equation (2), one would have expected a 25% effect. Also the dynamics of the $m_z$ remains very much the same: it compares similarly to the LL dynamics as we observed in Fig. 1. Definitely, there is no 25% effect. Hence, we confirm that the weak-field limit is given by the LL dynamics (1a) rather than by the LLG dynamics (1b).