# Peer review of "Landau-Lifshitz damping from Lindbladian dissipation in quantum magnets"

_SciPost Physics_

## Round 2 · Referee Report · Rembert Duine (Referee 1) · 2025-3-7

Strengths
1. mathematically technically clear
2. Mathematically valid
3. Reproducible
Weaknesses
1. Key approximation not physically motivated sufficiently
2. Overall motivation not sufficiently clear
Report
The work by Uhrig concerns a derivation of the Landau-Lifschitz equation, including damping in the Landau-Lifshitz form, from the Lindblad formalism. The key step of the author is to adapt the Lindblad operator, that enters the Master equation, to point in the direction of the local field – including external and internal exchange field - felt by the spin. As I understand it, this implies that within this assumption the spin exchanges angular momentum with the bath, and that this angular momentum is at each instant in time pointing in the direction of the field. After this approximation, and provided the external field is much lower than the exchange field, the author recovers the Landau-Lifshitz equation for the magnetization with the Landau-Lifshitz form of the damping. The requirement that the external field is much lower than the exchange field essentially leads to high-frequency dynamics due to the exchange field being damped out much quicker than the dynamics including the external field, such that the latter contribution survives.
The strong point of the paper is that the technical part of the paper is written in a clear way and all the steps mathematical steps are straightforward to follow.
In my opinion, there are, however, unclarities related to approximations and motivation that I find hard to grasp from the current version of the article:
Regarding the approximations: the author’s key assumption is that the Lindblad operator adapts instantaneously to the external field. The question is, how does the bath do this? Particularly, a phonon bath does not couple to an external magnetic field and there seems to be no physical mechanism for the bath to “know” which angular momentum to absorb.
Regarding motivation: the author insists that it should be possible to derive the Landau-Lifshitz equation from Lindbladian dynamics because “… the spins are quantum objects and cannot have their own relaxation independent from the general quantum theory.” To be honest, I do not fully understand this sentence. I will assume that it is intended to mean that the since Landau-Lifshitz equation describes spins, and that since spins are quantum objects of which the relaxation should be described by the Lindblad equation, it should be possible to derive the Landau-Lifshitz equation from the Lindblad equation. However, this motivation is unclear to me in the following sense: according to my understanding, the Landau-Lifshitz equation is intended to describe the low-frequency long wavelength dynamics of the magnetic order parameter far below the Curie temperature – so that amplitude fluctuations are irrelevant - and does not apply to an individual spin in a material. As such, the dynamics that the Landau-Lifshitz equation describes is inherently classical. Equations (5) and (6) suggest that an individual spin the material undergoes damped precession, even at the energy scales set by J, with a single damping parameter lambda. However, in any realistic material, I do not think this is the case. If one would experimentally probe a single spin at high frequencies, one would find that it decays in a very different way than the Landau-Lifshitz equation predicts. It is only the collective motion of the spins that at small frequencies and long wavelengths that should be described by the Landau-Lifshitz-equation. As a second, related, point, I do not think that the Landau-Lifshitz form of the damping is the appropriate one to focus on. The Gilbert damping form of the damping follows is in my opinion, more fundamental, as it does not depend on the field itself, but only on the frequency of the dynamics, similar to viscous friction forces in other systems. Finally, in the motivation, the author cites a paper, Ref. [15], where there is agreement between the Lindblad formalism and the Landau-Lifshitz equation. However, in this case the Lindblad formalism is applied to quantized excitations of the order parameter, magnons, rather than to microscopic spins.
In view of these unclarities in arguing for the key approximation and the overall motivation, I do not think the current form of the article meets SciPost’s criteria for publication.
Requested changes
1. Improve the discussion of why/how the Lindblad operator adapts to the field, indicate the regime of validity of this approximation, and discuss for which baths this approximation holds
2. Improve the motivation for this work. Argue convincingly why it is needed that the Landau-Lifshitz equation should follow from the Lindblad Master equations, and/or in which situations it should follow.
Recommendation
Reject
Author: Götz Uhrig on 2025-04-08 [id 5353]
(in reply to Report 1 by Rembert Duine on 2025-03-07)
Parts of the report are quoted to render it clear to which the reply refers.
"Weaknesses 1. Key approximation not physically motivated sufficiently" I provide further justification of the approximation here and in the revised version.
"2. Overall motivation not sufficiently clear" See below for my view on the overall motivation.
"The work by Uhrig ... latter contribution survives." By and large, I agree to the above summary, except for the last sentence since no high-frequency dynamics occur in the equation, neither in the Lindblad nor in the Landau-Lifshitz equation and thus, it is not damped out. Implicitly, however, there is some high-frequency dynamics going on in the bath which determines the Lindblad decay rate lambda.
"Regarding the approximations: the author’s key assumption is that the Lindblad operator adapts instantaneously to the external field. The question is, how does the bath do this? Particularly, a phonon bath does not couple to an external magnetic field and there seems to be no physical mechanism for the bath to “know” which angular momentum to absorb."
Thank you for this excellent question! Of course, the generic bath does not know about the orientation of a magnetic field. But this issue arises already in the first static case of a single spin. The answer is fundamental to thermodynamics: Boltzmann’s H theorem teaches us that the temporal evolution, at least after some coarse-graining of states, increases the total entropy. Hence, if there is a system in contact with a large bath at zero temperature the excess energy of in the system will flow into the bath and relax the system to its ground state in the long run. This is reflected in the Lindblad formalism in the condition that the Lindblad operator B_l increases the system’s energy. In this way, the bath “knows” the orientation of the magnetic field because only by aligning the spin to it the system’s energy is decreased, i.e., energy flows from the system into the bath incrementing its entropy.
The same holds for a slowly varying Hamiltonian if we assume that the bath is acting fast. The coupling to the bath will make the instantaneous energy in the system decrease.
The above reasoning has been included in the revised version for clarification.
"Regarding motivation: the author insists that it should be possible to derive the Landau-Lifshitz equation from Lindbladian dynamics because “… the spins are quantum objects and cannot have their own relaxation independent from the general quantum theory.” To be honest, I do not fully understand this sentence. I will assume that it is intended to mean that the since Landau-Lifshitz equation describes spins, and that since spins are quantum objects of which the relaxation should be described by the Lindblad equation, it should be possible to derive the Landau-Lifshitz equation from the Lindblad equation. " This is precisely the gist of the motivation, which I perhaps did not formulated optimally. I modified the formulation for improved clarity. I emphasize that the motivation in the paper is just that: a motivation for asking for a link between two formalisms out of scientific curiosity. The motivation is not meant to be a rigorous argument and I do not think that it would have to be since the following calculation realizes such a link.
"However, this motivation is unclear to me in the following sense: according to my understanding, the Landau-Lifshitz equation is intended to describe the low-frequency long wavelength dynamics of the magnetic order parameter far below the Curie temperature – so that amplitude fluctuations are irrelevant - and does not apply to an individual spin in a material. As such, the dynamics that the Landau-Lifshitz equation describes is inherently classical. " I agree as far as one restricts oneself to small deviations from equilibrium. But we address relaxation upon application of an external field and the system can have been manipulated before by coherent pulses to prepare non-equilibrium states. If it were in its unaltered ground state no relaxation would occur. But away from equilibrium it is a valid question to address fluctuations of the order parameter in the orientation and in the amplitude.
"Equations (5) and (6) suggest that an individual spin the material undergoes damped precession, even at the energy scales set by J, with a single damping parameter lambda. " Let me emphasize that the combination of equations (5) and (8) (formerly (6)) correspond to the self-consistent mean-field treatment of the order parameter, i.e., it describes the collective motion of the order parameter. Note that all local mean-field theories reduce an extended many body problem to an effective single site problem plus a self-consistency condition, see, e.g., the famous fermionic dynamic mean-field theory. Thus, the dynamics of the magnetization is not the dynamics of a single spin.
"However, in any realistic material, I do not think this is the case. If one would experimentally probe a single spin at high frequencies, one would find that it decays in a very different way than the Landau-Lifshitz equation predicts. It is only the collective motion of the spins that at small frequencies and long wavelengths that should be described by the Landau-Lifshitz-equation. " As pointed out above, it is precisely the dynamics of the order parameter which is described by the advocated equations in line with your argument.
"As a second, related, point, I do not think that the Landau-Lifshitz form of the damping is the appropriate one to focus on. The Gilbert damping form of the damping follows is in my opinion, more fundamental, as it does not depend on the field itself, but only on the frequency of the dynamics, similar to viscous friction forces in other systems. " The Referee knows more about LL and LLG treatments than I do. It is well possible that a link to the LLG equation would be even more desirable. Within the present line of argument, only the LL equation appeared although I looked for signs of LLG dynamics. Anyway, the found link between the Lindblad and Landau-Lifshitz is probably not yet the end of the story and further extensions are certainly possible.
"Finally, in the motivation, the author cites a paper, Ref. [15], where there is agreement between the Lindblad formalism and the Landau-Lifshitz equation. However, in this case the Lindblad formalism is applied to quantized excitations of the order parameter, magnons, rather than to microscopic spins." I quoted and summarized this nice paper of yours in the Introduction and I acknowledged that an agreement of both approaches was found, here and in the manuscript. If I am missing a point or mis-represent your results, please inform me so that I can improve on this point.
In my view, Ref. 15 goes beyond what my calculation is aiming at, but at the price of generality. The relaxation in Ref. 15 is dealt with not in a local mean-field theory, but on a spin wave level with magnons as damped harmonic oscillators. Thus, it is more microscopic, but less general: note that the agreement in Ref. 15 only occurs in leading order of the deviations from equilibrium and assuming special properties of the baths. In contrast, I only assume that the bath is fast and that the order parameter can point in any direction.
"In view of these unclarities in arguing for the key approximation and the overall motivation, I do not think the current form of the article meets SciPost’s criteria for publication." I hope to have provided sufficient explanations for the approximation. As for the motivation, I re-iterate that it is in essence scientific curiosity which indeed has led to an interesting link.
"Requested changes 1. Improve the discussion of why/how the Lindblad operator adapts to the field, indicate the regime of validity of this approximation, and discuss for which baths this approximation holds" This has been discussed here and in the revised manuscript.
"2. Improve the motivation for this work. Argue convincingly why it is needed that the Landau-Lifshitz equation should follow from the Lindblad Master equations, and/or in which situations it should follow." As for the motivation, it is in essence scientific curiosity which motivated me to search for a link between two seemingly very different formalisms. I do not think that this curiosity is a flaw.
Author: Götz Uhrig on 2025-04-08 [id 5352]
(in reply to Report 2 on 2025-03-10)Where necessary parts of the report are repeated and given in quotation marks.
"Strengths
1- Points out a basic feature of self-consistent use of Lindblad equation for single spin"
The formulation is slightly misleading because the self-consistency promotes the single-spin treatment to a local mean-field theory of an extended system
"One motivation is to provide a general approach for computing damping parameters from explicit descriptions of the reservoirs."
This sentence is not correct since the paper does not aim at an explicit description of the reservoirs, but establishes a fairly general link between a Lindbladian and the Landau-Lifshitz formalism of relaxation with a minimum of parameters.
"Although a basic calculation, this seems to be an interesting point."
Thank you for the interest in this result.
"However, the issue lies with motivating the equations and their use in self-consistent fashion. Here I concur with the first referee. As a result it remains unclear what precisely one can conclude from the (correct) analysis performed which aims to "fix a conceptual flaw". This leaves too many open ends to recommend publication in SciPost Physics. "
The modified version elucidates the arguments justifying the equations and removes some slightly inaccurate statements.
As for 1.
Equation (3) holds very generally; it is (almost) the text book version. In order to avoid confusion, I added explicit formulae for the Hamiltonians involved. It is true, that the statement on J was premature. It has been modified.
As for 2.
Even though it is very common to justify the Lindblad formalism in the weak-coupling limit its structure also holds for strong coupling. So, there is no need to impose a particularly small lambda. This is changed in the modified version. Moreover, the value J is indeed introduced later. Accordingly, a reference to J is postponed.
As for 3.
I agree that one has to assume this for the LL equation to be robust. But it is not the Lindblad formalism which requires it as clarified in the revised version.
As for 4.
"This does not justify it: it is just consistent with that approximation."
This is precisely what is meant: The approximation is justified a posteriori because the physics is consistent with its assumptions. The formulation has been adapted.
As for 5.
"- It is remarkable that in the case of saturation |m|⃗ = S the time scale 1/γ disappears completely and only the time scales 1/h0 and J/(γh0 ) survive."
I agree that this is truly remarkable making the presented finding an interesting and relevant result. As such it is not uncommon that certain fast energy scales drop out if one considers complex equations in certain limits.
"Unfortunately, this is not explored either but this should be easy with the numerics all set up. E.g. what special thing happens in the full self-consistent equations (5)+(6) for such initial lengths?"
I agree that the length change deserves investigation. Thus, the additional section 5 has been included discussing exemplary length changes on the basis of the extended formula.
As for 6.
"It is mentioned that the LL equation breaks down for λ>1. However, it is not clear or investigated whether the extended LL equation (7) is immune to any breakdown, even for λ<1, with the new term present for some initial magnitude of the |m|. "
The extended equation is not immune either because it becomes exactly the LL equation for saturated m. This is mentioned now in the revised version.
"Related: Is it possible that C=2S/|m|-1 changes sign during the dynamics, i.e., is it guaranteed that |m| < 2S with this term present?"
If |m| is the spin expectation value a sign change cannot happen because |m|<=S rigorously. In the (approximate) numerics an overshooting of |m| cannot be excluded fully. But no such phenomenon has occurred in my simulations. This is now mentioned in the revised version.
As for 7.
The formulation was indeed slightly inaccurate: The fact that the magnetic order can have arbitrary length which changes in time is clearly a sign of quantum mechanics because a classical vector would have a fixed and constant length. In this point the local mean-field theory goes beyond the classical description. But I agree that it is conceivable that for other models the local mean-field treatment is still equivalent to a classical treatment. It does not per se imply quantum behavior.
As for 8.
Thank you for pointing these issues out. All of them have been accounted for in the revised version.
Requested changes are carried out.

---

## Round 3 · Referee Report · Rembert Duine (Referee 1) · 2025-4-22

Strengths
Strengths: 1. mathematically technically clear 2. Mathematically valid 3. Reproducible
Weaknesses
- Key approximation not physically motivated sufficiently and convincingly
- Overall motivation not sufficiently clear, especially not from physics point of view
Report
In my previous report, I requested the following two changes: 1.Improve the discussion of why/how the Lindblad operator adapts to the field, indicate the regime of validity of this approximation, and discuss for which baths this approximation holds
- Improve the motivation for this work. Argue convincingly why it is needed that the Landau-Lifshitz equation should follow from the Lindblad Master equations, and/or in which situations it should follow. Regarding point 1: the author argues that the Lindblad operator adapts to the field because then the entropy of the system increases. However, to me this sounds like using a macroscopic law (2nd law of thermodynamics) to argue for a microscopic description (the Lindblad operator) that then is tweaked to the macroscopic requirements but is not necessarily rooted in reality. I would think that the macroscopic dynamics should follow from the microscopic description without the need for such finetuning. In physical magnets, there are well-known baths (phonons for insulators, electrons for metals), with well-known microscopic descriptions which do not a priori “know” of the second law of thermodynamics, and - for this case also relevant - do not involve the magnetic field. The resulting dynamics, when coupling the magnetic order parameter to the baths, is such that energy flows to the bath in certain limits and approximations, and that entropy increases. Typically this is because the bath has a large amount of degrees of freedom when compared to the system of interest, the precessing magnet. In conclusion, regarding point 1 I do not find the argument of why the Lindblad operator adapts to the field convincing. Regarding point 2: The author maintains in the manuscript that “If the damping of a magnetic systems can be described by the LL(G) equations, which is governed by a single relaxation rate, the latter should be derivable from Lindbladian dynamics. So far, however, Lindblad dynamics and LL(G) dynamics are not linked by a mathematically rigorous derivation, except that for special systems where their outcomes are the same [15] for small deviations from equilibrium. This situation is unsatisfactory: spins are quantum objects so that the Lindblad approach is applicable. “ Moreover, the author adds in the response that “As for the motivation, it is in essence scientific curiosity which motivated me to search for a link between two seemingly very different formalisms. I do not think that this curiosity is a flaw.” I do not have a problem with curiosity per se, but I do think that the results can be misleading. I reiterate my statement that the LLG is intended to describe the collective classical transverse dynamics of the magnetic order parameter. Modelling it as arising from single quantum spins by 1) ad hoc – see point 1 - choosing a Lindblad operator and 2) coupling the spins in a mean-field approximation is very likely to be far from the physical reality for reasons I mention above and in my previous report. However, not all readers may be aware of this and may take these results to built upon. One can see also in the overinterpretation of results from atomistic spin simulations. In such simulations, each individual spin is modelled classically by an LLG equation. Results from such simulations have their merit for prediction low-frequency dynamics, but should be treated with caution when the predictions involve time and length scales that involve the exchange interactions. Similarly, the authors’ equation (5) is basically an LLG which involves the exchange interactions after insertion of Eq. (8), but is not intended to work at those energy/time scales. In conclusion, from my understanding of the author’s motivation in the previous response letter, one could view the results in the article as a mathematical way to derive the LLG with Landau-Lifshitz damping from a Lindblad formalism. However, I do not think that physically the modelling is well-founded (point 1 above), or that physically a close link between the LLG and the Lindblad formalism as applied to a single spin should necessarily exist (point 2). I think that without elaborate and convincing discussions of both points I mention at the beginning of this report, the paper should not be published.
Requested changes
- Give elaborate discussion of range of validity of key assumption
- Give elaborate discussion of motivation, especially from physics point of view
Recommendation
Reject
Author: Götz Uhrig on 2025-05-12 [id 5472]
(in reply to Report 1 by Rembert Duine on 2025-04-22)Referee's statements are given in quotation marks.
"1. Improve the discussion of why/how the Lindblad operator adapts to the field, indicate the regime of validity of this approximation, and discuss for which baths this approximation holds."
The Referee finds the argument unconvincing that the relaxation as described by the Lindblad formalism by construction is a directed process towards thermal equilibrium. He calls this “tweaked” and “fine-tuned”.
The reproach that “fine-tuning” is needed must be refuted since no special values of the parameters need to be fixed to arrive at the result.
Moreover, the reproach of “tweaking” is puzzling by the Referee who employed the Lindblad formalism himself, see Ref. 15. Hence, he used the fact that the relaxation rates in this formalism fulfill a relation such that the quantum system approaches thermal equilibrium, see e.g., Sect. 3.3.2 in “Open Quantum Systems” by Breuer and Petruccione. This is what I used as well:
If the magnetic field were constant in time, nobody would doubt that the dissipation should be such that relaxation is towards thermal equilibrium. This is how the “bath knows about the magnetic field”. It “knows” about which jump operator increments or decrements the energy in the quantum system.
Thus, for a sufficiently slowly varying field it is natural to assume that the Lindblad operator favors alignment to that field. It is clearly pointed in the manuscript that this is an assumption.
"2. Improve the motivation for this work. Argue convincingly why it is needed that the Landau-Lifshitz equation should follow from the Lindblad Master equations, and/or in which situations it should follow."
"I do not have a problem with curiosity per se, but I do think that the results can be misleading. I reiterate my statement that the LLG is intended to describe the collective classical transverse dynamics of the magnetic order parameter. Modelling it as arising from single quantum spins by 1) ad hoc – see point 1 - choosing a Lindblad operator and 2) coupling the spins in a mean-field approximation is very likely to be far from the physical reality for reasons I mention above and in my previous report."
I re-emphasize that the single spin represents a generic spin in the many-spin ensembles. It allows one to compute the magnetic order parameter in the local mean-field approximation. This is very good approximation for ferromagnets in high dimensions and at low energies. Thus, the statement that this is far from physical reality is *not* justified.
"However, not all readers may be aware of this and may take these results to built upon. One can see also in the overinterpretation of results from atomistic spin simulations."
It may be that some readers misinterpret the results, but this cannot be a reason for rejection of the manuscript in which the ingredients, i.e., the necessary assumptions are clearly spelt out.
"In such simulations, each individual spin is modelled classically by an LLG equation. Results from such simulations have their merit for prediction low-frequency dynamics, but should be treated with caution when the predictions involve time and length scales that involve the exchange interactions. Similarly, the authors’ equation (5) is basically an LLG which involves the exchange interactions after insertion of Eq. (8), but is not intended to work at those energy/time scales."
Equation (5) is still far away from an LLG equation although it contains a double cross product. But is still linear in the magnetization in striking contrast to the LL or LLG equation. It is not used on the scale of the exchange coupling after inserting the mean-field self-consistency. I must emphasize that this high-energy scale has dropped out in Eq. (9), i.e., in the weak-field limit. Thus, we actually agree on the physics which is given by the low-energy dynamics of the order parameter. Hence, I cannot find Referee’s conclusion of rejection justified since the equations of the manuscript reproduce his view on the dynamics.
"In conclusion, from my understanding of the author’s motivation in the previous response letter, one could view the results in the article as a mathematical way to derive the LLG with Landau-Lifshitz damping from a Lindblad formalism. However, I do not think that physically the modelling is well-founded (point 1 above), or that physically a close link between the LLG and the Lindblad formalism as applied to a single spin should necessarily exist (point 2)."
This conclusion must be refuted in the strongest possible terms: point 1 is not justified because the use of the time-dependent Lindblad formalism is in-line with the standard directedness of any Lindblad approach. Point 2, see above, does not apply because it again refers a single spin ignoring the gist of the local mean-field theory.

---

## Round 3 · Author Response

to avoid inaccuracies. For details please see the comprehensive replies to the two reports.
This has improved the manuscript and strengthend the advocated link between two
well-established formalisms which even allowed to extend the Landau-Lifshitz equation.
For these reasons, I am resubmitting the manuscript herewith.

---

## Round 3 · List of Changes

1)
Paragraph added on the direction of relaxation and
how the bath can know about the direction of the (effective)
magnetic field.
2)
Reformulation of the motivation to look for a link
between the Landau-Lifshitz and the Lindblad formalism
3)
Justifications added for the equations where appropriate;
Hamiltonians specified and the discussion of the coupling J
postponed to where it has been introduced.
Stating that the approximation on the direction of the magnetization
to be almost parallel to the total field is justified only a posteriori.
4)
Pointing out that the structure of the Lindblad equation
does not depend on the weak-coupling limit.
5)
Instability of the extended equation is now related to the one of
the LL equations.
6)
Discussion of the value of C: no sign change occurs.
7)
Clarification that a local mean-field theory does not
per se imply quantum effects. But the possibility that the
length of the order parameter changes does represent
a quantum effect.
8)
Sect. 5 with Fig. 2 is added to discuss generic solution
for the length changes of the order parameter due to relaxation.
9)
All smaller typos and spotted formulation errors are removed.

---

## Editorial Decision

unknown